# TRIP13 localizes to synapsed chromosomes and functions as a dosage-sensitive regulator of meiosis

Jessica Y Chotiner[1], N Adrian Leu[1], Fang Yang[1], Isabella G Cossu[1], Yongjuan Guan[1,2], Huijuan Lin[1], P Jeremy Wang[1]*

[1]Department of Biomedical Sciences, University of Pennsylvania School of Veterinary Medicine, Philadelphia, United States; [2]College of Life Sciences, Capital Normal University, Beijing, China

*For correspondence: pwang@vet.upenn.edu

**Abstract** Meiotic progression requires coordinated assembly and disassembly of protein complexes involved in chromosome synapsis and meiotic recombination. Mouse TRIP13 and its ortholog Pch2 are instrumental in remodeling HORMA domain proteins. HORMAD proteins are associated with unsynapsed chromosome axes but depleted from the synaptonemal complex (SC) of synapsed homologs. Here we report that TRIP13 localizes to the synapsed SC in early pachytene spermatocytes and to telomeres throughout meiotic prophase I. Loss of TRIP13 leads to meiotic arrest and thus sterility in both sexes. *Trip13*-null meiocytes exhibit abnormal persistence of HORMAD1 and HOMRAD2 on synapsed SC and chromosome asynapsis that preferentially affects XY and centromeric ends. These major phenotypes are consistent with reported phenotypes of *Trip13* hypomorph alleles. *Trip13* heterozygous mice exhibit meiotic defects that are less severe than the *Trip13*-null mice, showing that TRIP13 is a dosage-sensitive regulator of meiosis. Localization of TRIP13 to the synapsed SC is independent of SC axial element proteins such as REC8 and SYCP2/SYCP3. Terminal FLAG-tagged TRIP13 proteins are functional and recapitulate the localization of native TRIP13 to SC and telomeres. Therefore, the evolutionarily conserved localization of TRIP13/Pch2 to the synapsed chromosomes provides an explanation for dissociation of HORMA domain proteins upon synapsis in diverse organisms.

## eLife assessment

This **important** study defined the physiological function of a conserved meiosis factor during murine spermatogenesis. The genetic and cellular biological evidence supporting the conclusion is **convincing**. This work will be of broad interest to cell biologists, geneticists, and reproductive biologists.

## Introduction

Meiosis is a specialized cell division program that generates haploid gametes from diploid germ cells. During meiotic prophase I, homologous chromosomes undergo paring, synapsis, and recombination (*Handel and Schimenti, 2010*; *Zickler and Kleckner, 2015*). Chromosome synapsis requires the formation of the synaptonemal complex (SC). The SC is a proteinaceous structure comprising two lateral/axial elements, transverse filaments, and a central element (*Page and Hawley, 2004*). Meiotic recombination begins with the generation of DNA double-strand breaks (DSBs) and ends with the crossover formation through repair of DSBs (*Hunter, 2015*). Chromosome synapsis and meiotic recombination are interdependent in many species, including yeast and mouse. Meiotic recombination

not only increases genetic diversity in gametes at the organism level but also is essential for faithful chromosome segregation during the first meiotic cell division. Therefore, defects in meiosis are the leading causes of aneuploidy, birth defect, infertility, and pregnancy loss.

The progression and completion of synapsis and recombination are monitored by a surveillance mechanism called meiotic checkpoint (*Roeder and Bailis, 2000*). Meiotic checkpoint proteins were extensively studied in yeast. Many meiotic processes are highly conserved, and studies of meiosis in yeast have been foundational for our understanding of mammalian meiosis. In yeast, Pch2, an AAA+ATPase, is a checkpoint protein because it is necessary for pachytene arrest in the absence of the SC protein Zip1 or the meiosis-specific DSB repair protein Dmc1 (*Herruzo et al., 2021*; *San-Segundo and Roeder, 1999*). Pch2 is a hexameric ring ATPase and remodels the HORMA (Hop1, Rev7, and MAD2) domain protein Hop1 in a nucleotide-dependent manner (*Chen et al., 2014*). While Pch2 is mainly located in the nucleolus, Pch2 also localizes to the synapsed SC as foci at the pachytene stage and is essential for removing Hop1 from chromosome axes (*Chen et al., 2014*; *Joshi et al., 2009*; *San-Segundo and Roeder, 1999*). Mechanistically, Pch2 promotes phosphorylation of Hop1, which activates the Mek1 kinase and the subsequent checkpoint cascade (*Herruzo et al., 2016*; *Raina and Vader, 2020*).

TRIP13, the mammalian ortholog of yeast Pch2, is essential for the completion of meiotic recombination in mouse (*Li and Schimenti, 2007*; *Roig et al., 2010*). Mutations in human *TRIP13* predispose to Wilms tumor in children or cause infertility in women (*Yost et al., 2017*; *Zhang et al., 2020*). HORMAD1 and HORMAD2, mammalian meiosis-specific HORMA domain proteins, localize to chromosome axis at leptotene and zygotene stages but are depleted from synapsed chromosomes, except the largely unsynapsed XY chromosomes at the pachytene stage (*Cossu et al., 2024*; *Fukuda et al., 2010*; *Xu et al., 2019*). TRIP13 facilitates removal of HORMAD proteins from synapsed chromosome axis (*Roig et al., 2010*; *Wojtasz et al., 2009*). SKP1, an essential component of the SCF (Skp1-Cullin 1-F-box protein) ubiquitin E3 ligase, is required not only for depleting HORMAD1 and HORMAD2 from synapsed chromosome axes but also for restricting the accumulation of HORMAD proteins on unsynapsed axes (*Guan et al., 2020*; *Guan et al., 2022*). Therefore, reorganization of HORMA domain proteins on meiotic chromosome axes is regulated by both TRIP13 and SKP1. As a conserved mechanism, Pch2/TRIP13 remodels HORMA domain proteins through engagement of their N-terminal regions (*Prince and Martinez-Perez, 2022*). MAD2, a spindle assembly checkpoint HORMA domain protein, exists in two fold states: closed and open (*Luo et al., 2004*; *Sironi et al., 2002*). In the closed state, the so-called safety belt of the HORMA domain wraps around the HORMAD protein-binding motif 'closure motif' in the partner protein and thus traps the partner. In the open state, TRIP13 engagement changes conformation of the HORMA domain, resulting in foldback of the safety belt on the closure motif-binding region and thus release of its partner protein. The safety belt mechanism turns out to be a shared feature of HORMA domain proteins (*Kim et al., 2014*; *West et al., 2019*; *West et al., 2018*; *Ye et al., 2017*).

Unlike yeast Pch2, TRIP13 does not appear to function in the meiotic checkpoint in mouse (*Pacheco et al., 2015*). Interestingly, HORMAD2 functions as a meiotic checkpoint protein for surveillance of meiotic defects in female meiosis (*Kogo et al., 2012a*; *Rinaldi et al., 2017*; *Wojtasz et al., 2012*). HORMAD2-deficient males are sterile but females are fertile. HORMAD2 deficiency rescues the oocyte loss in *Spo11*-null or *Trip13* mutant females but not in *Dmc1*-null females, suggesting that HORMAD2 is a component of the meiotic checkpoint in females. HORMAD1 is required for fertility in both sexes (*Kogo et al., 2012b*; *Shin et al., 2010*; *Stanzione et al., 2016*). Both HORMAD1 and HORMAD2 localize prominently to the XY chromosome axes in pachytene spermatocytes and regulate meiotic sex chromatin inactivation (*Kogo et al., 2012a*; *Shin et al., 2010*; *Wojtasz et al., 2012*).

Studies of mice with hypomorphic *Trip13* mutations show that TRIP13 is required for meiotic progression (*Li and Schimenti, 2007*; *Roig et al., 2010*). Two different hypomorphic *Trip13* mouse lines have been studied: one 'moderate' allele that exhibited defective meiotic recombination and one 'severe' allele that displayed defects in meiotic recombination and chromosome synapsis. The variance in phenotype between the hypomorphic mouse lines indicates that even partial depletion of *Trip13* can interrupt meiosis. We sought to further investigate the meiotic function of TRIP13. By immunofluorescence, we found that TRIP13 localized to the SC in early pachytene spermatocytes and to telomeres throughout meiotic prophase I. The TRIP13 localization pattern on meiotic chromosomes was further confirmed by the analysis of FLAG-tagged TRIP13 in two knockin mouse lines.

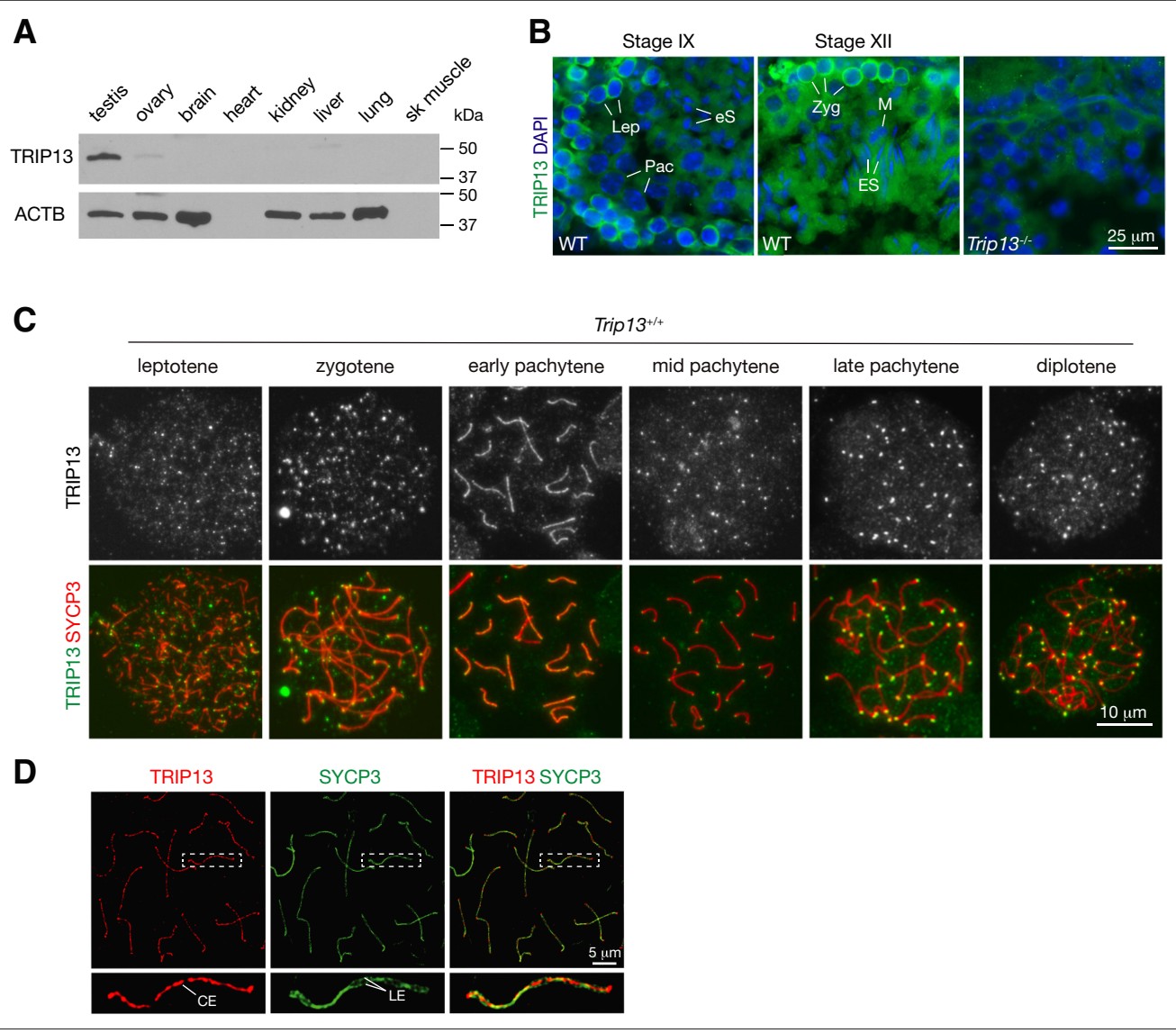

**Figure 1.** TRIP13 localizes to the synaptonemal complex and telomeres in spermatocytes. (**A**) Western blot analysis of TRIP13 in adult mouse tissues. Heart and skeletal muscle lack ACTB. (**B**) Immunofluorescence of TRIP13 in sections of 3-month-old wild type and *Trip13-/-* testes. Lep, leptotene; Zyg, zygotene; Pac, pachytene; eS, elongating spermatids; ES, elongated spermatids. (**C**) Immunofluorescence of TRIP13 in spread nuclei of spermatocytes from wild type P20 testes. (**D**) Super-resolution localization of TRIP13 to the central element (CE) but not lateral element (LE) of the synaptonemal complex at early pachytene stage. The enlarged view of the boxed chromosome is shown at the bottom.

The online version of this article includes the following source data and figure supplement(s) for figure 1:

**Source data 1.** Original files of the full raw unedited western blots in *Figure 1A*.

**Figure supplement 1.** Immunofluorescent analysis of TRIP13 in spread nuclei of oocytes from embryonic day 16.5 (E16.5) and E18.5 female embryos.

To determine the effect of complete loss of TRIP13, we characterized *Trip13*-null mice. While the *Trip13*-null phenotype was largely comparable to the phenotype of the severe *Trip13* hypomorphic allele, we found that TRIP13 is a dosage-sensitive regulator of meiosis. Finally, our results show that the localization of TRIP13 to the SC is independent of axial element proteins.

## Results

### TRIP13 localizes to meiotic chromosomes in prophase I spermatocytes

We assessed the expression of TRIP13 in a panel of adult mouse tissues. TRIP13 was primarily expressed in testis but detected at low levels in ovary and liver (*Figure 1A*). In the testis, TRIP13 was prominent in the cytoplasm of primary spermatocytes, especially leptotene and zygotene cells (*Figure 1B*). Given the expression of TRIP13 in spermatocytes, we examined its localization on meiotic chromosomes by immunostaining of surface spread nuclei of prophase I spermatocytes (*Figure 1C*). Previously, TRIP13 was reported to localize to telomeres in spermatocytes (*Gómez-H et al., 2019*). Indeed, we found that TRIP13 localized to telomeres from leptotene to diplotene cells (*Figure 1C*). Strikingly, we found that, in addition to telomeres, TRIP13 localized to the SC in early pachytene spermatocytes but disappeared from the SC in the mid to late pachytene spermatocytes (*Figure 1C*). The SC consists of two lateral elements and one central element. The central region comprises the transverse filament protein SYCP1 and central element proteins, which appear upon synapsis. Given the novel localization of TRIP13 to the SC, we performed confocal immunofluorescent microscopy with super-resolution deconvolution to determine the location of TRIP13 within the SC. In contrast to the lateral element marker SYCP3, TRIP13 localized to the SC central region (*Figure 1D*). We examined the localization of TRIP13 in oocytes. TRIP13 localized strongly to meiotic telomeres in all stages of prophase I, but not as filaments on the SC (*Figure 1—figure supplement 1*). These findings are consistent with the genetic requirement of TRIP13 in meiosis in both sexes (*Li and Schimenti, 2007*; *Roig et al., 2010*).

### Global loss of *Trip13* causes meiotic arrest

Two previous studies utilizing two *Trip13* hypomorphic (gene trap) alleles, one moderate and one severe, revealed a role for TRIP13 in meiotic recombination (*Li and Schimenti, 2007*; *Roig et al., 2010*). However, while the *Trip13* moderate allele showed mostly intact chromosomal synapsis, the severe allele displayed chromosomal unsynapsis preferentially at chromosome ends (*Li and Schimenti, 2007*; *Roig et al., 2010*). Upon close examination, we also found unsynapsed ends in spermatocytes with the moderate *Trip13* mutant (*Guan et al., 2020*). To rigorously ascertain the role of TRIP13 in meiosis, we generated *Trip13*-null mutants using frozen sperm from *Trip13*$^{+/-}$ males with a knockout allele from the International Mouse Phenotyping Consortium. The mouse *Trip13* gene consists of 13 exons. This new *Trip13* mutant allele harbors a deletion of a 24 kb region including all 13 exons and thus is expected to be null. We verified this deletion by PCR and sequencing. Interbreeding of *Trip13*$^{+/-}$ mice produced fewer *Trip13*$^{-/-}$ offspring than expected: *Trip13*$^{+/+}$, 80; *Trip13*$^{+/-}$, 220; *Trip13*$^{-/-}$, 51 ($\chi^2$ = 27, p=0.0001). The body weight of 2–3-month-old males was not significantly different between wild type (24.3 ± 2.8 g, n = 5) and *Trip13*$^{-/-}$ mice (22.8 ± 1.7 g, n = 5, p=0.3, Student's *t*-test). The *Trip13*$^{-/-}$ testis was much smaller than *Trip13*$^{+/-}$ testis (*Figure 2A*). Western blot analysis showed that TRIP13 was present in reduced abundance in *Trip13*$^{+/-}$ testis and not detected in *Trip13*$^{-/-}$ testis, demonstrating that this mutant allele is null (*Figure 2B*). The abundance of SYCP3, a component of the SC, was also reduced in *Trip13*$^{-/-}$ testis, suggesting a partial depletion of meiotic cells (*Figure 2B*). The testis weight of adult *Trip13*$^{-/-}$ males was reduced by 74% in comparison with the wild type males (*Figure 2C*). Adult *Trip13*$^{-/-}$ males lacked sperm in the epididymis (*Figure 2D*). Histological analysis showed that while spermatocytes in all stages of meiosis were present in adult *Trip13*$^{+/+}$ and *Trip13*$^{+/-}$ testes, *Trip13*$^{-/-}$ testis displayed complete meiotic arrest, evidenced by the presence of early spermatocytes and a lack of secondary spermatocytes, round spermatids, and mature sperm (*Figure 2E*). *Trip13*$^{-/-}$ females were also sterile. The adult *Trip13*$^{-/-}$ ovary was very small and showed a complete loss of oocytes (*Figure 2—figure supplement 1*). These observations are similar to the meiotic arrest phenotype observed in the hypomorphic *Trip13* mouse mutants (*Li and Schimenti, 2007*; *Roig et al., 2010*).

### Defects in chromosomal synapsis in *Trip13*-deficient spermatocytes

We investigated the effect of *Trip13* deficiency on meiotic progression. The transverse filament/central element protein SYCP1 and the lateral element protein SYCP3 were used to determine the stage of prophase I spermatocytes. While the P20 wild type testis contained spermatocytes from leptotene through diplotene stages, about half of the *Trip13*$^{-/-}$ spermatocytes were in the early pachytene stage and no cells were found at later stages of prophase I, showing meiotic blockade at the early pachytene stage in *Trip13*$^{-/-}$ testes (*Figure 3A*). The early pachytene-like spermatocytes

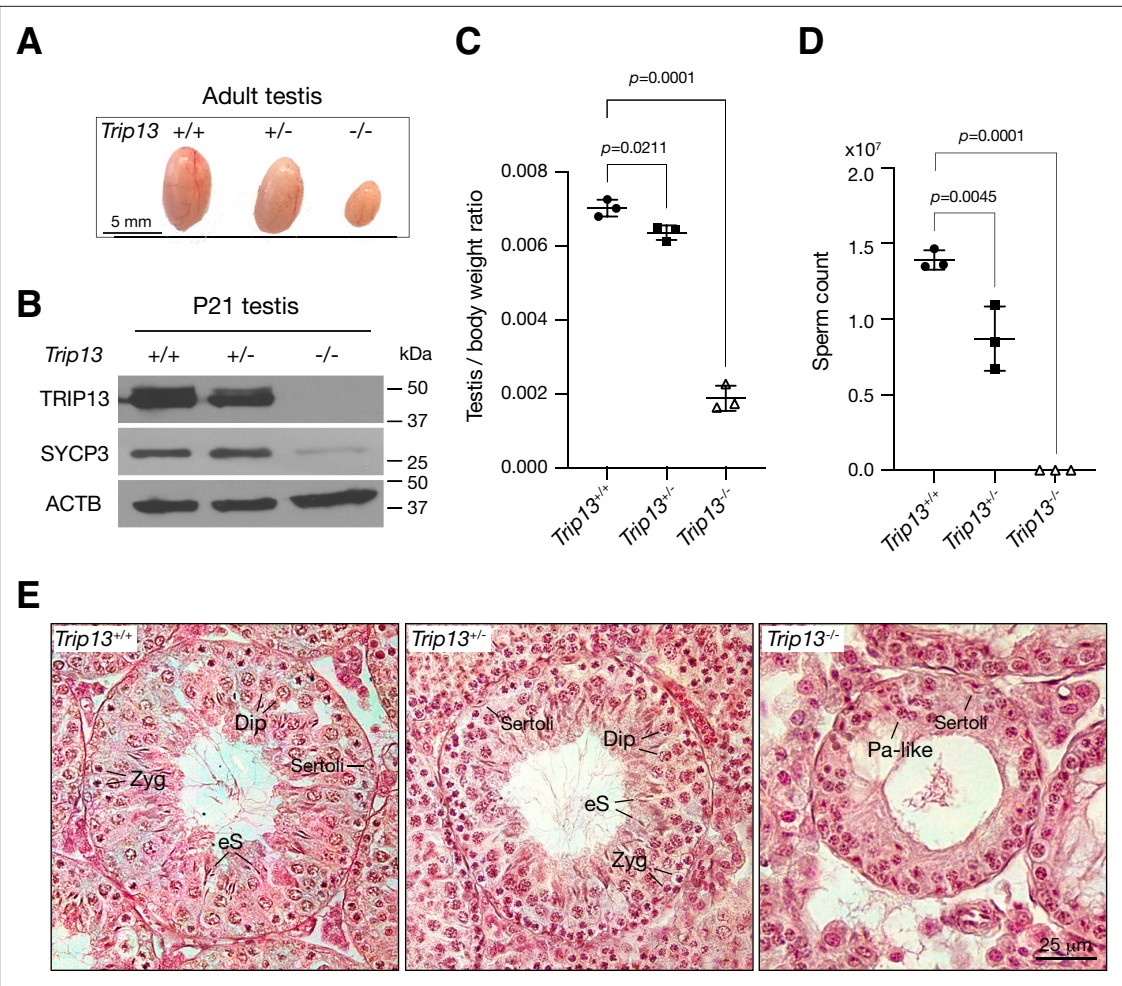

**Figure 2.** Loss of *Trip13* leads to meiotic arrest in males. (**A**) Image of testes from 2- to 3-month-old mice. (**B**) Western blot analysis of TRIP13 in P21 testes. SYCP3 serves as a meiosis-specific marker. ACTB serves as a loading control. (**C**) Testis to body weight ratio of 2–3-month-old mice. n = 3 males. Statistics, one-way ANOVA. (**D**) Sperm count of 2–3-month-old *Trip13*+/+ and *Trip13*+/- males. n = 3 males. Statistics, one-way ANOVA. (**E**) Histological analysis of 2-month-old testes. Sertoli, Sertoli cell; Zyg, zygotene; Pa-like, pachytene-like; Dip, diplotene; eS, elongating spermatids.

The online version of this article includes the following source data and figure supplement(s) for figure 2:

**Source data 1.** Original files of the full raw unedited western blots in *Figure 2B*.

**Figure supplement 1.** Histological analysis of ovaries from adult (8-week) wild type and *Trip13*-/- females.

from *Trip13*-/- testes contained unsynapsed chromosomal ends (*Figure 3B*). Mouse centromeres are telocentric. Co-staining with the centromere marker CREST showed that 94% of unsynapsed ends were centromeric ends (*Figure 4A*). While XY chromosomes were synapsed at the pseudoautosomal regions and thus were connected in wild type and *Trip13*+/- pachytene spermatocytes, they were separate in *Trip13*-/- pachytene-like spermatocytes (*Figure 3B and C*). We found that SYCE1, a component of the SC central element, localized to the synapsed SC but not to unsynapsed regions in *Trip13*-/- pachytene-like cells (*Figure 3C*). Confocal microscopy with super-resolution deconvolution confirmed that many homologous chromosomes in *Trip13*-/- spermatocytes had split ends and some had regions of interstitial asynapsis (*Figure 3D*). We quantified these meiotic defects in spermatocytes: chromosomal asynapsis (*Figure 3E*), the number of asynapsed ends (*Figure 3F*), and the extent of XY asynapsis (*Figure 3G*). Previous studies also reported synaptic defects in spermatocytes from *Trip13* hypomorph mutants (*Li and Schimenti, 2007*; *Roig et al., 2010*). Here we found that global loss of *Trip13* caused similar defects in autosomal synapsis but a more severe defect in sex chromosome synapsis.

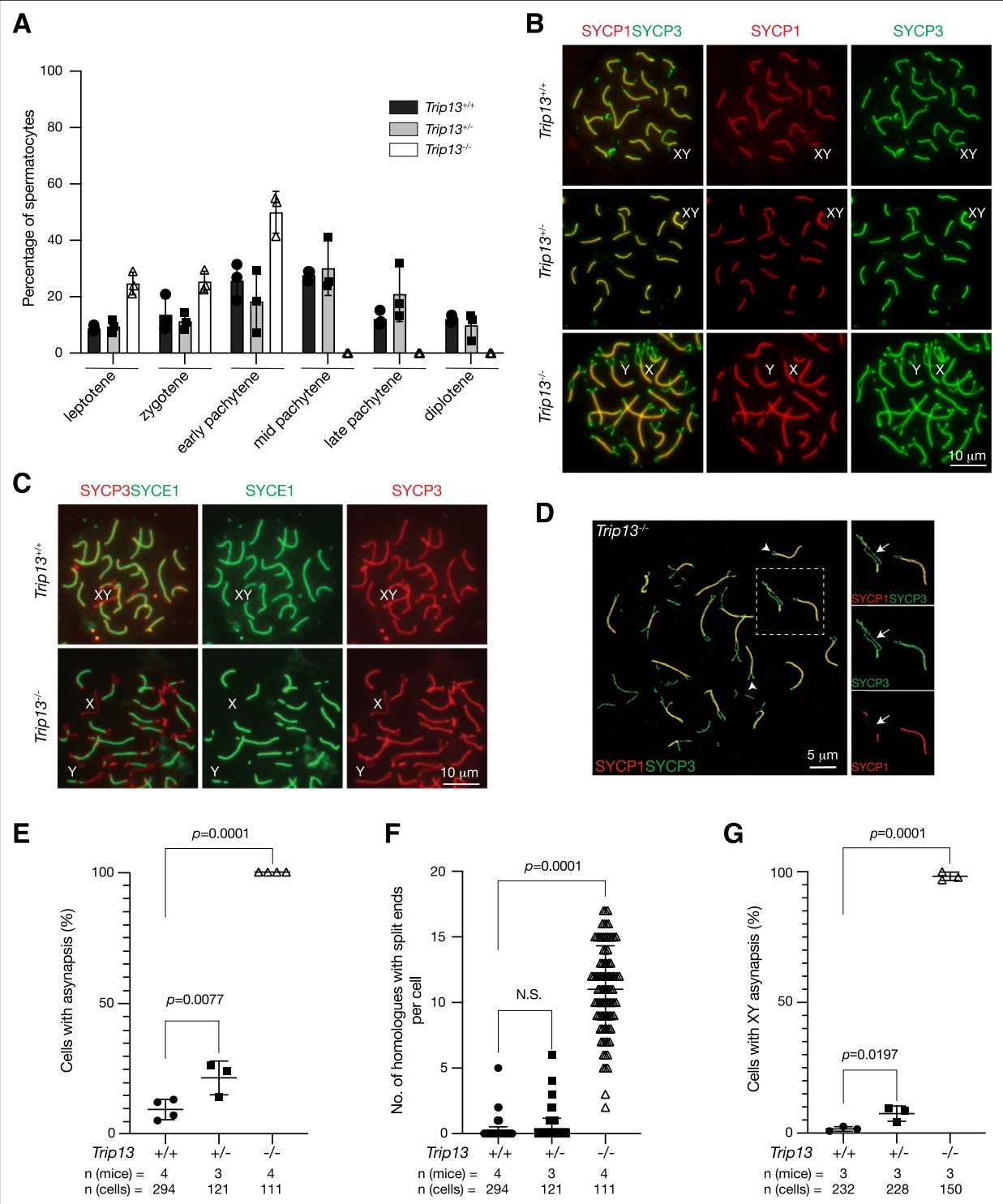

Figure 3. *Trip13* is required for chromosomal synapsis in males. (**A**) Composition of prophase I spermatocytes in P20 testes. Three males per genotype were analyzed by nuclear spread analysis. Total number of spermatocytes counted: *Trip13+/+*, 1038 cells; *Trip13+/-*, 803 cells; *Trip13-/-*, 406 cells. (**B**) Immunofluorescence of SYCP1 and SYCP3 in spread nuclei of pachytene spermatocytes from P20 testes. (**C**) Immunofluorescence of SYCE1 and SYCP3 in spread nuclei of pachytene spermatocytes from P20 testes. (**D**) Super-resolution confocal microscopy of a *Trip13-/-* spermatocyte from P20 testis. Immunostaining was performed for SYCP1 and SYCP3. Arrowheads indicate end asynapsis. Arrow indicates interstitial asynapsis. (**E**) Percentage of early pachytene cells from P19-20 testes with asynapsed chromosomes across three genotypes. (**F**) Number of homologous chromosomes with end asynapsis per cell in P19-20 testes. (**G**) Percentage of early pachytene cells from P20 testes with asynapsed XY chromosomes per mouse. The p-values are indicated in graphs. Statistics (**E–G**), one-way ANOVA.

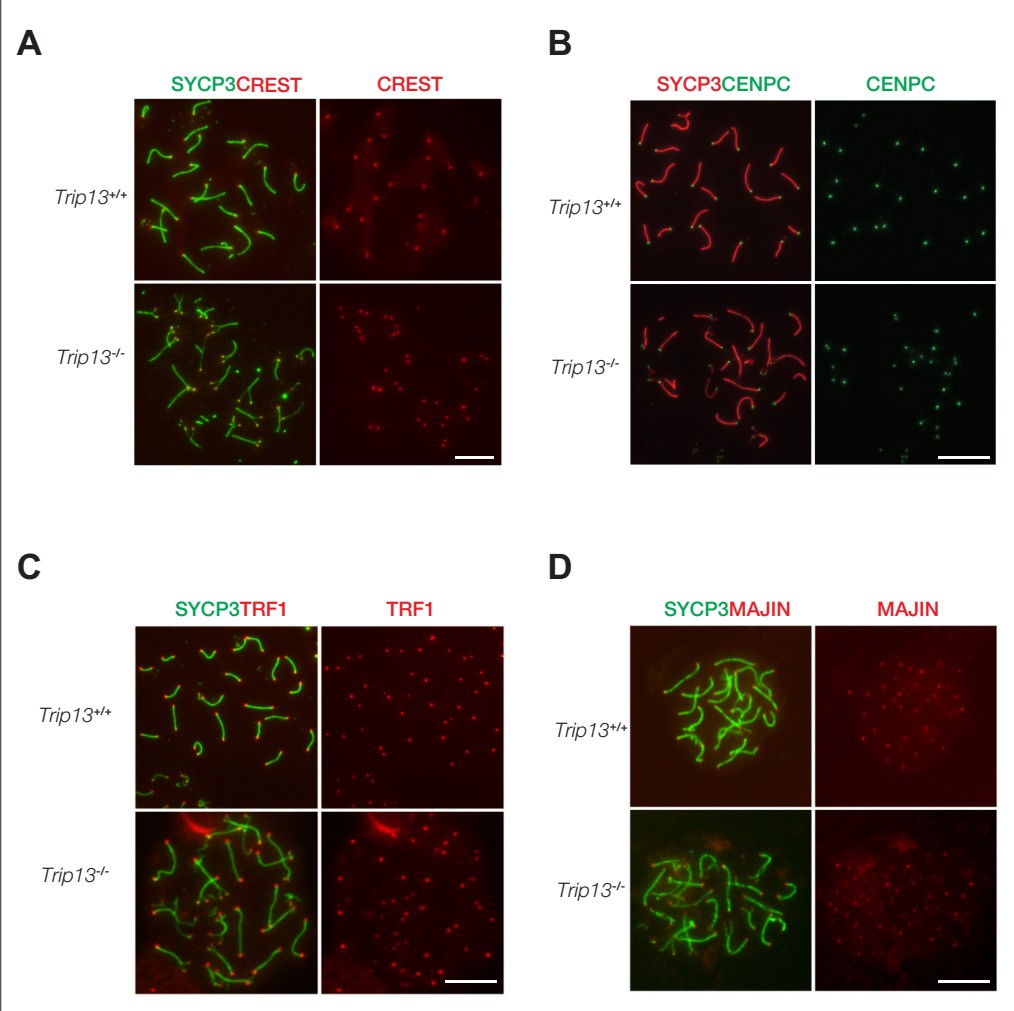

**Figure 4.** Normal localization of centromere and telomere markers in *Trip13*-deficient spermatocytes from juvenile mice. (**A–D**) Immunofluorescent analysis of centromere and telomere markers in *Trip13*+/+ and *Trip13*-/- pachytene spermatocytes: CREST (**A**), CENPC (**B**), TRF1 (**C**), and MAJIN (**D**). SYCP3 labels the lateral elements of the synaptonemal complex. Scale bars, 10 μm.

The online version of this article includes the following figure supplement(s) for figure 4:

**Figure supplement 1.** Immunofluorescent analysis of REC114 in spread nuclei of spermatocytes from P28 wild type (*Trip13*+/+) and *Trip13*-/- testes.

## TRIP13 is a dosage-sensitive regulator of meiosis

Although *Trip13*+/- males were fertile, their testis weight and sperm count were significantly reduced in comparison with wild type (*Figure 2C and D*). The percentage of spermatocytes with defects in synapsis was higher in *Trip13*+/- males than wild type (*Figure 3E*). In addition, the percentage of spermatocytes with asynapsed XY chromosomes was significantly higher in *Trip13*+/- males than wild type (*Figure 3G*). As defects in synapsis can activate the meiotic checkpoint and thus apoptosis of affected spermatocytes, the increased synaptic defects likely caused the decrease in sperm output in *Trip13*+/- males. These results demonstrate that TRIP13 regulates meiosis in a dosage-dependent manner.

## Telomere and centromere proteins localize normally in *Trip13*-deficient spermatocytes

TRIP13 localizes to telomeres and loss of TRIP13 causes peri-centromeric/telomeric asynapsis. Therefore, we asked whether telomere or centromere proteins were affected in *Trip13*-/- spermatocytes. Chromosome spreads were immunostained for two centromere markers, CREST and CENPC

(*Figure 4A and B*). CENPC is essential for recruiting kinetochore proteins to the centromere (*Klare et al., 2015*; *Kwon et al., 2007*). CREST and CENPC still localized to centromeres in *Trip13*⁻/⁻ spermatocytes. Because many *Trip13*⁻/⁻ spermatocytes had split ends, two CREST or two CENPC foci were observed at the split ends. To assess the telomere, spermatocytes were immunostained for TRF1 and MAJIN (*Figure 4C and D*). In meiotic cells, telomeres contain canonical telomere proteins such as TRF1/TRF2 and a meiosis-specific complex (MAJIN, TERB1, TERB2, and SUN1) (*Shibuya et al., 2015*). TRF1 is a telomere protein expressed in both somatic and germ cells (*Karlseder et al., 2003*; *Long et al., 2017*; *Shibuya et al., 2014*). MAJIN is a component of the meiosis-specific telomere complex that is important for telomere attachment to the inner nuclear membrane (*Shibuya et al., 2015*). Both TRF1 and MAJIN localized to telomeres in *Trip13*⁻/⁻ spermatocytes (*Figure 4C and D*). As expected, two TRF1 or two MAJIN foci were observed at split ends of chromosomes in *Trip13*⁻/⁻ spermatocytes (*Figure 4C and D*). ANKRD31 and REC114 interact with each other and both localize to the pseudo-autosomal region (PAR) of XY chromosomes in early pachytene spermatocytes to ensure XY recombination (*Boekhout et al., 2019*; *Papanikos et al., 2019*). In addition to two foci on autosomes, REC114 localized as one focus on the PAR in wild type pachytene spermatocytes. REC114 still formed foci (one per chromosome) on the unsynapsed X and Y chromosomes in *Trip13*⁻/⁻ pachytene-like spermatocytes (*Figure 4—figure supplement 1*). Taken together, these results suggest that TRIP13 is not required for recruitment of these centromere or telomere proteins.

## TRIP13 is required to evict HORMAD1 and HORMAD2 from synapsed autosomes

In wild type meiotic cells, HORMAD1 and HORMAD2 localize to unsynapsed and desynapsed chromosomes (*Fukuda et al., 2010*; *Kogo et al., 2012a*; *Shin et al., 2010*; *Wojtasz et al., 2012*). Thus, HORMAD1 and HORMAD2 localized to the largely unsynapsed XY but not to synapsed autosomal SCs in wild type cells (*Figure 5A and B*). TRIP13 is essential for removing meiotic HORMADs from the chromosome axes, a function that is conserved in yeast, worms, and mammals (*Wojtasz et al., 2009*). We confirmed that HORMAD1 and HORMAD2 remained on the synapsed autosomes in *Trip13*⁻/⁻ pachytene cells (*Figure 5A and B*). HORMAD1 and HORMAD2 localize to the interior of the lateral elements in the SC (*Xu et al., 2019*). Confocal microscopy with super-resolution deconvolution revealed that HORMAD1 and HORMAD2 localized to the lateral elements of the synapsed autosomes in *Trip13*⁻/⁻ spermatocytes (*Figure 5C and D*). These results were consistent with accumulation of HORMAD1/2 in *Trip13* hypomorphic mutant spermatocytes (*Roig et al., 2010*; *Wojtasz et al., 2009*). Our analysis of *Trip13*-null mutant confirmed that TRIP13 is essential for HORMAD1/2 removal from the synapsed SCs.

## Localization of TRIP13 to SC is independent of axial element components

TRIP13 localizes to the SC in early pachytene spermatocytes. We sought to address what proteins might recruit TRIP13 to the SC. We examined the role of HORMAD1, REC8, SYCP2, and SKP1 in TRIP13 localization in spermatocytes using respective knockout mice (*Figure 6*). We first examined the localization of TRIP13 in *Hormad1*⁻/⁻ spermatocytes. *Hormad1*⁻/⁻ spermatocytes exhibit limited synapsis (*Daniel et al., 2011*; *Kogo et al., 2012b*; *Shin et al., 2010*). In both wild type and *Hormad1*⁻/⁻ cells, TRIP13 localized to the synapsed regions and to the telomeres (*Figure 6A*). This result shows that HORMAD1 is not required for localization of TRIP13 to the SC.

REC8, a meiosis-specific cohesin, promotes synapsis between homologs and inhibits synapsis between sister chromatids (*Xu et al., 2005*). In the absence of REC8, synapsis occurs between sister chromatids rather than homologous chromosomes as previously shown by super-resolution imaging (*Guan et al., 2020*). In *Rec8*⁻/⁻ spermatocytes, TRIP13 localized to synapsed sister chromatids (*Figure 6B*). To further probe TRIP13 recruitment to the SC, a mouse line expressing a truncated SYCP2 was used (*Yang et al., 2006*). SYCP2 and SYCP3 are essential components of the axial/lateral elements of the SC and interact with each other. The mutant used in this study (referred to as *Sycp2*⁻/⁻) expresses a truncated SYCP2 protein that lacks the C-terminal coiled-coil domain necessary for binding to SYCP3. Thus, SYCP3 failed to localize to the axial elements and formed aggregates in *Sycp2*⁻/⁻ spermatocytes (*Figure 6C*). Nevertheless, *Sycp2*⁻/⁻ spermatocytes formed short stretches of synapsis as previously demonstrated by electron microscopy and SYCP1 immunofluorescence (*Yang*

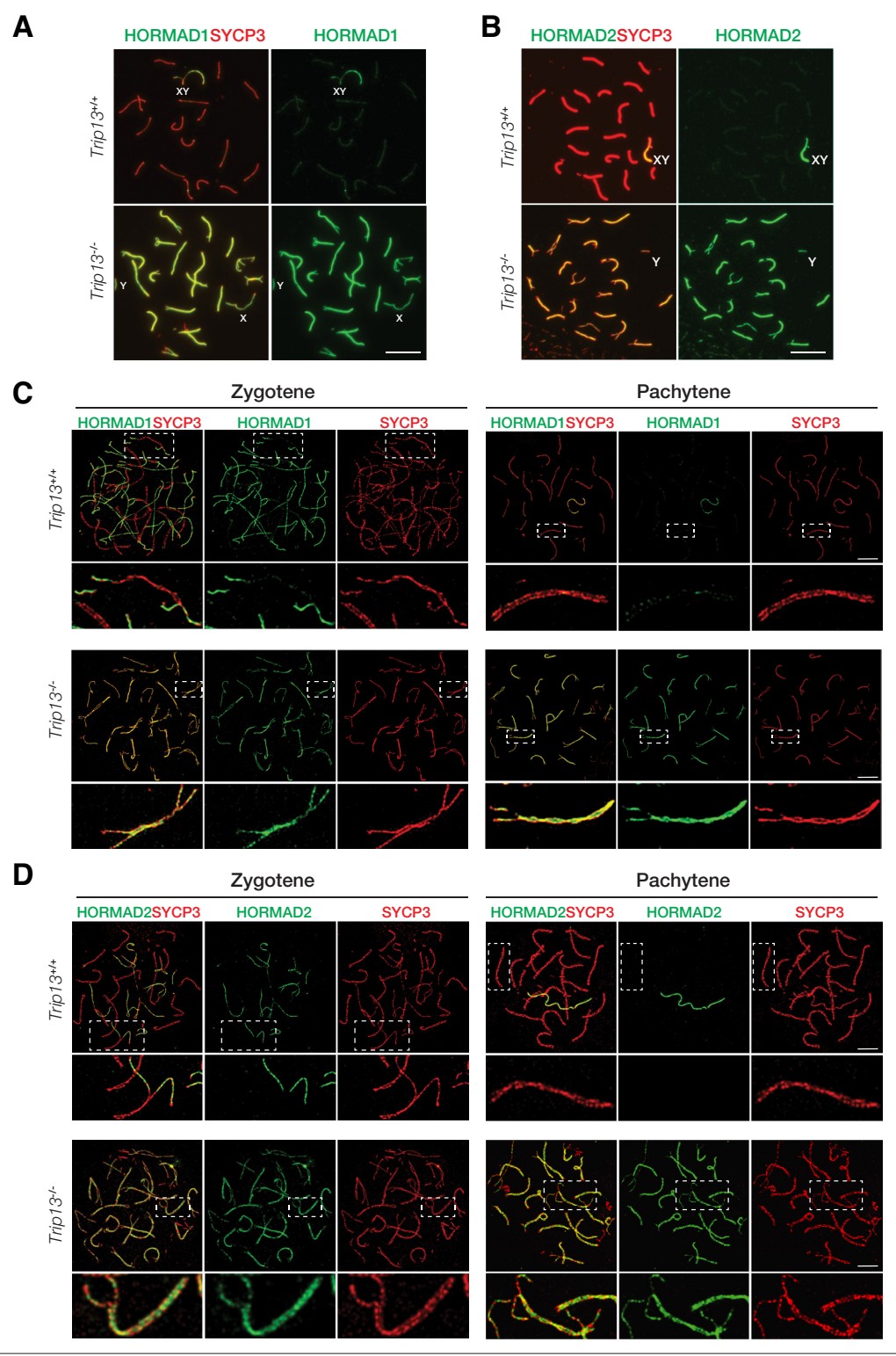

**Figure 5.** HORMAD1 and HORMAD2 accumulate on the lateral elements of synapsed autosomes in *Trip13*^-/-^ spermatocytes from juvenile (P19-21) mice. (**A, B**) Immunofluorescence of HORMAD1 (**A**) and HORMAD2 (**B**) in pachytene spermatocytes. (**C, D**) Super-resolution imaging of HORMAD1 and HORMAD2 in zygotene and pachytene spermatocytes. Enlarged views of the boxed chromosomes are shown below. Scale bars: 10 μm (**A, B**), 5 μm (**C, D**).

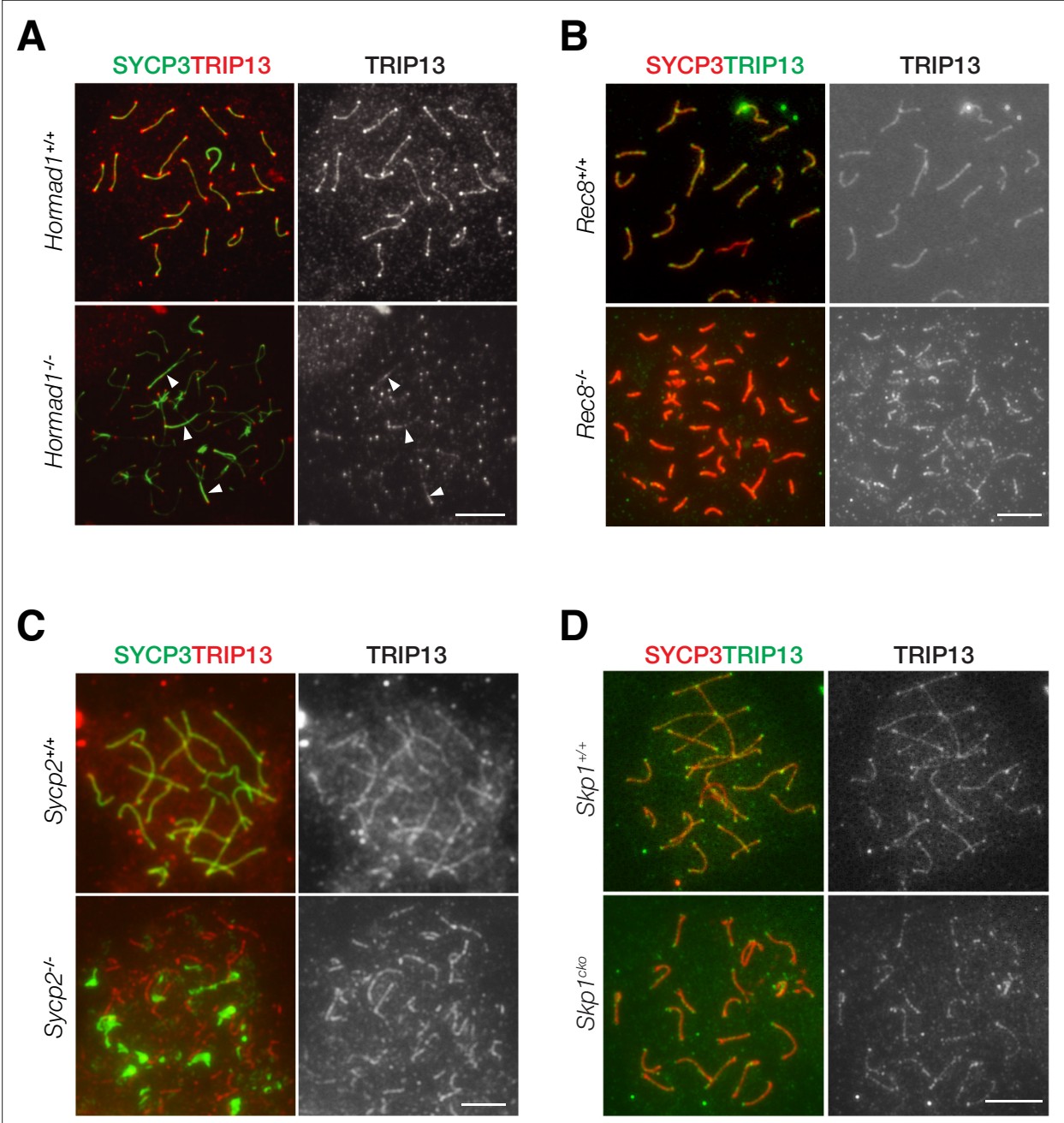

**Figure 6.** Localization of TRIP13 to the synaptonemal complex (SC) is independent of individual axial element components. (**A**) Immunofluorescent analysis of TRIP13 in *Hormad1⁻/⁻* spermatocytes from 2-month-old mice. (**B**) Immunofluorescent analysis of TRIP13 in *Rec8⁻/⁻* spermatocytes from 2-month-old mice. (**C**) Immunofluorescent analysis of TRIP13 in *Sycp2⁻/⁻* spermatocytes from 2-month-old mice. (**D**) Immunofluorescent analysis of TRIP13 in *Skp1ᶜᴷᴼ* spermatocytes from 2-month-old mice. Scale bars, 10 μm.

*et al., 2006*). TRIP13 localized as filaments in *Sycp2⁻/⁻* spermatocytes, strongly suggesting that its localization is independent of SYCP3 and the C-terminus of SYCP2 (*Figure 6C*).

Previous work has shown that SKP1, a key component of the SKP1, Cullin, F-box (SCF) complex E3 ligase, is important for HORMAD removal and chromosomal synapsis (*Guan et al., 2020*; *Guan et al., 2022*). SKP1 localizes to synapsed regions in meiotic germ cells and specifically to the lateral elements in the SC (*Guan et al., 2020*). We found that TRIP13 still localized to the synapsed regions in *Skp1ᶜᴷᴼ* (conditional knockout) spermatocytes (*Figure 6D*). Taken together, these results show that the SC localization of TRIP13 is independent of HORMAD1, REC8, SYCP2, SYCP3, and SKP1, which

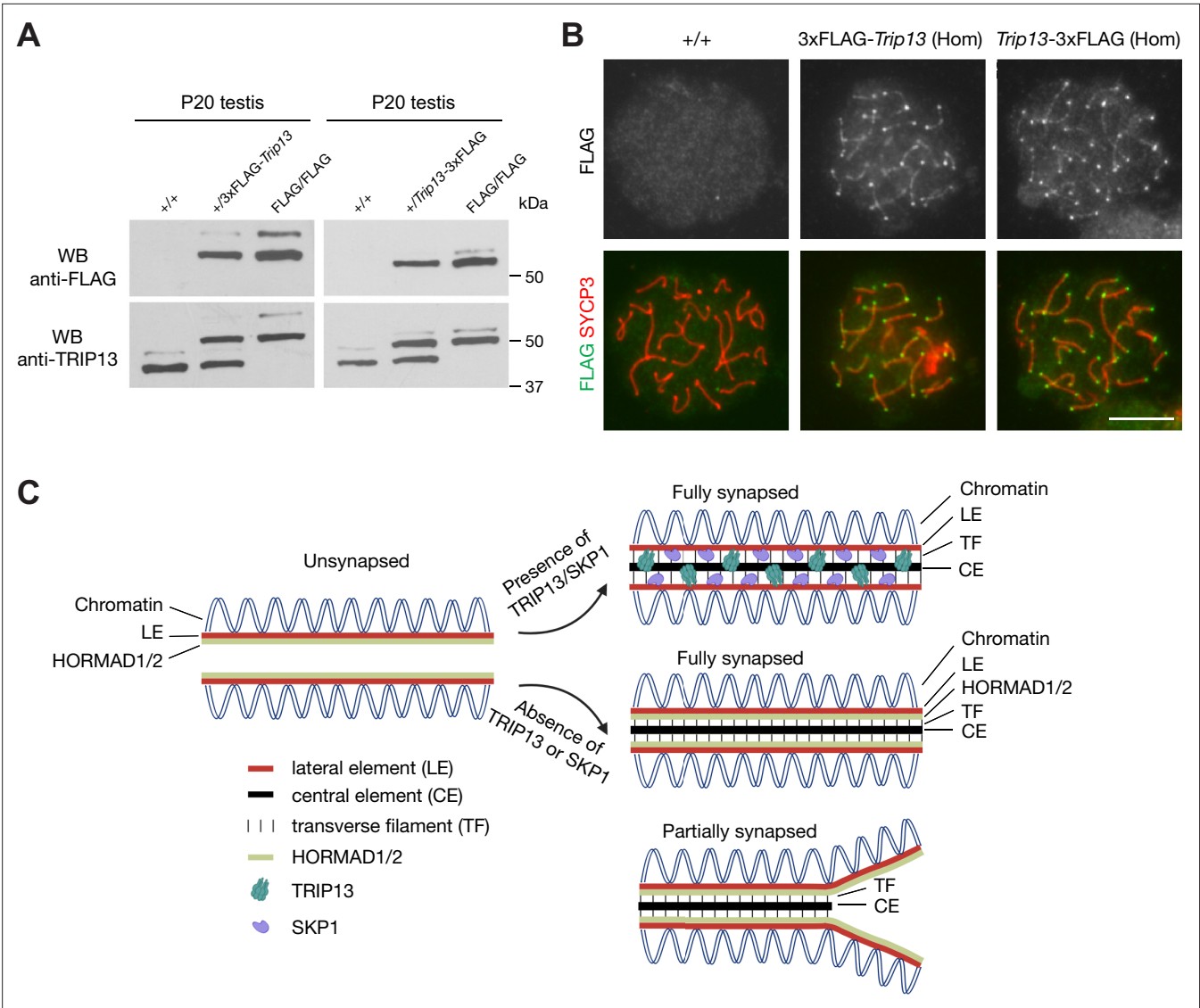

**Figure 7.** FLAG-tagged TRIP13 proteins localize correctly and are functional. (**A**) Western blot analysis of tagged and untagged TRIP13 proteins in testes from P20 wild type (no tag), heterozygous-tagged, and homozygous-tagged males. (**B**) Immunofluorescence of FLAG-tagged TRIP13 in pachytene spermatocytes from P20 homozygous testes. N-terminal tag, 3×FLAG-*Trip13*; C-terminal tag, *Trip13*–3×FLAG. Scale bar, 10 μm. (**C**) A schematic illustration of TRIP13, SKP1, HORMAD1/2, and the synaptonemal complex. Relative locations of TRIP13 and SKP1 within the synaptonemal complex (SC) are depicted. HORAMD1/2 are retained in synapsed regions in *Trip13*-deficient or *Skp1*-deficient spermatocytes.

The online version of this article includes the following source data and figure supplement(s) for figure 7:

**Source data 1.** Original files of the full raw unedited western blots in *Figure 7B*.

**Figure supplement 1.** Generation of two *Trip13* FLAG-tagged mouse lines.

all localize to the SC lateral elements. Such a finding is consistent with the localization of TRIP13 to the SC central region (*Figure 1D*).

## FLAG-tagged TRIP13 proteins are functional

In order to investigate the mechanism of TRIP13 recruitment to the SC, we generated 3×FLAG-TRIP13 (N-terminal tag) and TRIP13–3×FLAG (C-terminal tag) mice through the CRISPR/Cas9-medidated genome editing approach (*Figure 7—figure supplement 1*). Two different types of tagged mice were generated in case one fusion protein was not functional. Both alleles were transmitted through the germline of the founder mice. Western blot showed that TRIP13 existed in two isoforms in wild type (non-tagged) testis (*Figure 7A*). The major TRIP13 isoform was 50 kDa. The minor isoform was slightly

**Table 1.** List of proteins from testis identified by co-immunoprecipitation and mass spectrometry.

| Protein | Number of peptides in anti-FLAG IP | | | MW (kDa) |
| --- | --- | --- | --- | --- |
| | 3×FLAG-TRIP13 testis | TRIP13–3×FLAG testis | Wild type testis | |
| TRIP13 | 28 | 36 | 4 | 51 |
| HORMAD2 | 21 | 10 | 7 | 39 |
| DDX46 | 32 | 6 | 0 | 117 |
| PUF60 | 24 | 14 | 0 | 59 |
| RBM25 | 19 | 3 | 2 | 100 |
| RBM39 | 20 | 10 | 1 | 59 |
| U2AF1 | 11 | 4 | 1 | 28 |
| U2AF2 | 15 | 9 | 3 | 54 |
| SRSF11 | 9 | 4 | 0 | 53 |
| SPATA5 | 26 | 27 | 7 | 97 |
| AP3D1 | 17 | 6 | 2 | 135 |
| PRPF40A | 16 | 3 | 0 | 108 |
| UGP2 | 15 | 3 | 0 | 57 |

larger than 50 kDa. Western blot analysis showed that the FLAG-tagged TRIP13 fusion proteins were present in both FLAG-tagged testes. Both FLAG-tagged TRIP13 fusion proteins also existed in two isoforms in the homozygous testes, suggesting that they corresponded to the two wild type isoforms. The 3×FLAG-TRIP13 proteins were apparently slightly larger, possibly due to the linker in the N-terminally tagged proteins. The nature and physiological significance of these two isoforms were not clear, but could be due to alternative splicing or post-translational modification. Immunofluorescence analysis using anti-FLAG antibody showed that the FLAG-tagged TRIP13 proteins, like wild type TRIP13, localized to both telomeres and SC in pachytene spermatocytes from both FLAG/FLAG homozygous testes (*Figure 7B*). Importantly, both N- and C-terminally tagged homozygous mice were fertile. These results demonstrate that both N- and C-terminal tagged TRIP13 proteins localize normally and are functional.

To identify TRIP13-associated proteins in testis, we performed immunoprecipitation (IP) using 3×FLAG-TRIP13, TRIP13–3×FLAG, and wild type (no tag) testicular protein extracts with anti-FLAG monoclonal antibody. The immunoprecipitated proteins were eluted with FLAG peptides and subjected to mass spectrometry for protein identification. As expected, TRIP13 had more peptides in tagged TRIP13 IP than wild type (*Table 1*). HORMAD2, a known TRIP13 substrate protein, was also enriched in tagged TRIP13 IP (*Table 1*). Intriguingly, a large number of RNA-binding proteins involved in RNA splicing were highly enriched in tagged TRIP13 IP: DDX46, PUF60, RBM25, RBM39, U2AF1, U2AF2, and SRSF11 (*Table 1*). The biological relevance of these RNA-binding proteins to TRIP13 function remains unknown. We cannot exclude the possibility that the identification of these RNA splicing factors could be immunoprecipitation artifacts.

## Discussion

TRIP13 and its ortholog Pch2 play an evolutionarily conserved role in the depletion of meiosis-specific HORMA domain proteins from the SC in many species, including yeast, plant, and mouse. In meiocytes, HORMA domain proteins are associated with unsynapsed chromosome axes and become depleted from the SC upon synapsis (*Figure 7C*). In mice with *Trip13* hypomorphic gene trap alleles, HORMAD1 and HORMAD2 persist on the synapsed SCs at the pachytene stage (*Wojtasz et al., 2009*). In this study, we confirmed the abnormal persistence of HORMAD1 and HORMAD2 in *Trip13*-null spermatocytes (*Figure 7C*). In budding yeast, deficiency of Pch2 leads to the accumulation of Hop1 on the SC (*Joshi et al., 2009*; *Subramanian et al., 2016*). In *Arabidopsis*, Pch2 remodels the HORMA domain protein ASY1 (*Balboni et al., 2020*; *Yang et al., 2020*). Pch2 has been shown to localize to synapsed

SC in budding yeast (*Joshi et al., 2009*), *Caenorhabditis elegans* (*Deshong et al., 2014*; *Russo et al., 2023*), *Arabidopsis* (*Lambing et al., 2015*), and rice (*Miao et al., 2013*), but not in mice. In mouse meiocytes, TRIP13 was known to localize to telomeres (*Gómez-H et al., 2019*). In addition to telomeres, we find that TRIP13 localizes to the SC in early pachytene spermatocytes in mice. In particular, TRIP13 localizes as a single filament between the SC lateral elements. Therefore, TRIP13/Pch2 and HORMA domain proteins have mutually exclusive localization patterns on the meiotic chromosome axes, providing an explanation for the depletion of HORMA domain proteins from the synapsed chromosomes in diverse organisms.

The transverse filaments (TFs) are essential for the recruitment of Pch2 to the SC. In a budding yeast Zip1 (encoding the TF component) allele with four amino acid substitutions, Pch2 is absent from the SC (*Subramanian et al., 2016*). In *C. elegans*, the TF component SYP-1 is essential for localization of Pch-2 on the paired chromosomes (*Deshong et al., 2014*). In rice, CRC1 (Pch2 ortholog) and CEP1 (TF protein) localize to the SC in an inter-dependent manner (*Miao et al., 2013*). In *Arabidopsis*, ZYP1 (TF component) recruits Pch2 to the SC (*Yang et al., 2022*). In the yeast two-hybrid assay, rice CRC1 interacts with CEP1, *Arabidopsis* ZYP1 interacts with Pch2, and mouse SYCP1 (TF protein) interacts with TRIP13 (*Miao et al., 2013*; *Yang et al., 2022*). Unlike in other species, loss of SYCP1 (TF protein) in mouse leads to a complete failure in chromosomal synapsis and thus its role in the recruitment of TRIP13 to the synapsed SC cannot be directly addressed (*de Vries et al., 2005*). Our findings on the localization of mouse TRIP13 to the SC on the sister chromatids in *Rec8*-deficient spermatocytes and the short SC stretches in *Sycp2* mutant spermatocytes demonstrate that TRIP13 recruitment is independent of SC axial element proteins REC8 and SYCP2 but support the possibility that SYCP1 could recruit TRIP13 to the synapsed SC. These findings lead to the following model: the TF protein recruits Pch2/TRIP13 to the SC upon synapsis, which in turn evicts HORMA domain proteins from the synapsed regions (*Figure 7C*).

Both TRIP13 and SKP1 are required for the removal of HORMA domain proteins from the synapsed SC (*Guan et al., 2020*; *Wojtasz et al., 2009*). However, the relationship of TRIP13 and SKP1 is unknown. SKP1 localizes to the synapsed SC more extensively than TRIP13. While TRIP13 is only detected on the synapsed SC in early pachytene spermatocytes (*Figure 1*), SKP1 localizes to the synapsed SC in zygotene, all stages of pachytene, and diplotene spermatocytes (*Guan et al., 2020*). The localization of TRIP13 and SKP1 within the SC is different: TRIP13 is on CE/TF regions but SKP1 on LEs. While the global level of TRIP13 is reduced in *Skp1*-deficient testes, the localization of TRIP13 and SKP1 to the synapsed SC is independent (*Figure 6D*; *Guan et al., 2020*). The SCF ubiquitin E3 ligase complex targets HORMAD1 for ubiquitination and degradation in transfected HEK293T cells (*Guan et al., 2022*). Loss of TRIP13 leads to the accumulation of HORMAD proteins only to the synapsed SC in pachytene cells. In contrast, depletion of SKP1 causes accumulation of HORMA domain proteins, particularly HORMAD1, on both unsynapsed and synapsed chromosome axes from leptotene through diplotene meiocytes. Therefore, TRIP13 and SKP1 might regulate different pools of HORMAD proteins in meiocytes (*Figure 7C*).

In many organisms, including mouse and human, centromeres are the last regions to synapse (*Bisig et al., 2012*; *Brown et al., 2005*; *Qiao et al., 2012*). The reason for this is unknown, but it might be related to the necessity to suppress deleterious non-homologous recombination at centromere and pericentromeric regions, which consist of minor and major satellite repeats, respectively. The centromere was proposed to exert an inhibitory effect on synapsis in human cells (*Brown et al., 2005*). The inhibition needs to be relieved to achieve full synapsis since unsynapsis at centromeres is expected to trigger the meiotic checkpoint, leading to meiotic arrest. To date, only two mouse mutants (*Skp1* and *Trip13*) exhibit unsynapsis preferentially at the centromeric end in pachytene-like cells. In addition, only these two mouse mutants display abnormal accumulation of HORMAD proteins on the synapsed SC. Intriguingly, TRIP13 localizes to telomeres in both spermatocytes and oocytes. The centromere is close to one of the telomeres in mouse. Thus, the preferential centromeric end asynapsis in *Trip13* or *Skp1*-deficient meiocytes could be related to the abnormal persistence of HORMAD proteins. However, the underlying molecular mechanism warrants further investigation.

We find that TRIP13 is a dosage-dependent regulator of meiosis. The *Trip13*[+/-] mice displayed reduced testis weight, reduced sperm count, and meiotic defects but these defects were less severe than the *Trip13*[-/-] mice. The disease phenotypes in humans also appear to be influenced by the *TRIP13* dosage (*Yost et al., 2017*; *Zhang et al., 2020*). The dosage-dependent phenotypic variation has

been reported in other mouse meiotic mutants such as *Tex11*, *Meiob*, and *Rnf212*. TEX11 localizes to meiotic chromosomes as foci and regulates crossover formation and chromosome synapsis (*Yang et al., 2008*). The TEX11 protein levels correlate with genome-wide recombination rates in mice (*Yang et al., 2015*). MEIOB, a meiosis-specific ssDNA-binding protein, is essential for meiotic recombination (*Luo et al., 2013*; *Souquet et al., 2013*). In mice with different combinations of *Meiob* alleles, MEIOB protein levels directly correlate with the severity of meiotic defects (*Guo et al., 2020*). Sequence variants in human RNF212 are associated with variations in genome-wide recombination rates (*Chowdhury et al., 2009*; *Kong et al., 2008*). In mouse, RNF212 forms foci on meiotic chromosomes and regulates crossover formation in a dosage-dependent manner (*Reynolds et al., 2013*). Therefore, dosage sensitivity appears to be a common feature of many regulators of meiosis.

# Materials and methods

## Key resources table

| Reagent type (species) or resource | Designation | Source or reference | Identifiers | Additional information |
|---|---|---|---|---|
| Gene (*Mus musculus*) | *Trip13* | GenBank | Gene ID: 69716 | |
| Genetic reagent (*M. musculus*) | *Trip13*tm1.1(KOMP)Vlcg/JMmucd | MMRCC | MMRRC_050223-UCD | |
| Genetic reagent (*M. musculus*) | *Hormad1* knockout | PMID:21079677; *Shin et al., 2010* | | Rajkovic lab |
| Genetic reagent (*M. musculus*) | *Sycp2* knockout | PMID:16717126; *Yang et al., 2006* | | Wang lab |
| Genetic reagent (*M. musculus*) | *Rec8* knockout | PMID:32232159; *Guan et al., 2020* | | Wang lab |
| Genetic reagent (*M. musculus*) | 3×FLAG-*Trip13* | This paper | | Wang Lab |
| Genetic reagent (*M. musculus*) | *Trip13*-3×FLAG | This paper | | Wang Lab |
| Antibody | Anti-ACTB (mouse monoclonal) | Sigma | Cat# A5441, RRID:AB_476744 | WB (1:2000) |
| Antibody | Anti-centromere (CREST) (human polyclonal) | Antibodies Incorporated | Cat# 15-234, RRID:AB_2687472 | IF (1:500) |
| Antibody | Anti-SYCP1 (rabbit polyclonal) | Abcam | Cat# ab15090, RRID:AB_301636 | IF (1:300) |
| Antibody | Anti-SYCP2 (guinea pig polyclonal) | PMID:16717126 | Custom made | IF (1:150) |
| Antibody | Anti-SYCP3 (mouse monoclonal) | Abcam | Cat# ab97672, RRID:AB_10678841 | IF (1:500) |
| Antibody | Anti-SYCP3 (rabbit polyclonal) | ProteinTech Group | Cat# 23024-1-AP, RRID:AB_11232426 | IF (1:500), WB (1:2000) |
| Antibody | Anti-HORMAD1 (rabbit polyclonal) | ProteinTech Group | Cat# 13917-1-AP, RRID:AB_2120844 | IF (1:300) |
| Antibody | Anti-HORMAD2 (rabbit polyclonal) | PMID:19851446; *Wojtasz et al., 2009* | A gift from Toth lab | IF (1:500) |
| Antibody | Anti-REC114 (rabbit polyclonal) | PMID:31003867; *Boekhout et al., 2019* | A gift from Keeney Lab | IF (1:100) |
| Antibody | Anti-REC8 (rabbit polyclonal) | Custom made | A gift from Mengcheng Luo lab | IF (1:200) |
| Antibody | Anti-FLAG (mouse monoclonal) | Sigma | Cat# F3165, RRID:AB_259529 | IF (1:300), WB (1:4000) |
| Antibody | Anti-SKP1 (rabbit polyclonal) | Cell Signaling | Cat# 12248S, RRID:AB_2754993 | IF (1:200), WB (1:1000) |
| Antibody | Anti-TRIP13 (rabbit polyclonal) | ProteinTech Group | Cat# 19602-1-AP | IF (1:150), WB (1:1000) |

*Continued on next page*

| Reagent type (species) or resource | Designation | Source or reference | Identifiers | Additional information |
|---|---|---|---|---|
| Antibody | Anti-SYCE1 (rabbit polyclonal) | Custom made | A gift from Mengcheng Luo lab | IF (1:200), WB (1:1000) |
| Antibody | Anti-SYCP1 (mouse monoclonal) | Custom made | A gift from Christer Hoog lab | IF (1:200), WB (1:1000) |
| Antibody | Anti-CENP-C (rabbit polyclonal) | PMID:25533956; *Kim et al., 2015* | A gift from Y. Watanabe lab | IF (1:500) |
| Antibody | Anti-TRF1 (mouse monoclonal) | Sigma | Cat# T1948 | IF (1:100) |
| Antibody | Anti-MAJIN (guinea pig polyclonal) | This paper | Custom made | IF (1:100) |

## Materials availability statement

Newly created materials (3×FLAG-*Trip13* and *Trip13*-3×FLAG mice) are freely available, subject to standard institutional material transfer agreement.

## Mouse strains

The cryopreserved sperm from *Trip13*$^{+/-}$ males were obtained from the Mutant Mouse Resource and Research Center at UC Davis (*Trip13*$^{tm1.1(KOMP)Vlcg/JMmucd}$, MMRRC_050223-UCD). Genotyping of wild type and knockout *Trip13* alleles was performed by separate PCR reactions using tail genomic DNA. Other mutant mouse lines used in this study were previously generated: *Hormad1*, *Sycp2*, and *Rec8* (*Guan et al., 2020*; *Shin et al., 2010*; *Yang et al., 2006*). Genotyping PCR primer sequences are listed in *Table 2*.

## Generation of 3×FLAG-*Trip13* and *Trip13*-3×FLAG knockin mouse strains

The 3×FLAG-knockin mouse strains were generated using the CRISPR-Cas9-mediated genome editing approach. To generate N-terminally tagged knockin mice, one single-guide RNA (sgRNA) was designed to target the first exon of the mouse *Trip13* (*Figure 7—figure supplement 1A*). An ssDNA repair template was created using the sequence surrounding the start codon as 5′ and 3′ homology arms. The knockin template itself included the sequence for the 3×FLAG epitope and a three-residue linker sequence. The C-terminally tagged strain was generated similarly (*Figure 7—figure supplement 1B*). The guide RNA targeted exon 13 containing the stop codon. The ssDNA template contained the FLAG-encoding sequences and homology sequences flanking the insertion site. The sgRNA sequences and ssDNA template sequences are shown in *Table 2*.

For the sgRNA, the oligo was phosphorylated, annealed, and cloned to PX330 plasmid (Addgene, Waterton, MA). After in vitro transcription with the MEGAshortscript T7 Kit (AM1354, Invitrogen) and purification with the MEGAclear Transcription Clean-Up Kit (AM1908, Invitrogen), a mixture of Cas9 mRNA (100 ng/μl; Trilink, Cat# L-7206 + 0.5 μl of the sgRNA [35 ng/μl] + 100 ng/ul of ssDNA template) was prepared and injected into zygotes. The injected zygotes were cultured in KSOM medium at 37°C in a 5% $CO_2$ incubator until the two-cell stage. The two-cell embryos were transferred into oviducts of 0.5-day post-coitum pseudopregnant ICR foster mothers. Founder mice were bred to wild type mice to obtain germline transmission. The N-terminal 3×FLAG allele and the C-terminal 3×FLAG allele were PCR amplified and sequenced to confirm the insertion. PCR genotyping primers are listed in *Table 2*.

## Production of anti-MAJIN antibodies

The short isoform of mouse MAJIN (amino acids 1–124; XM_036161650.1 and XP_036017543.1) was expressed as a 6xHis-MAJIN recombinant protein in *Escherichia coli* using the pQE-30 vector. The recombinant protein was affinity purified with the Ni-NTA agarose. Two guinea pigs were immunized at Cocalico Biologicals Inc (Reamstown, PA), resulting in two antisera (UP-GP140 and UP-GP141). Both anti-sera were used for immunofluorescence of nuclear spreads of spermatocytes.

**Table 2.** Sequences of genotyping PCR primers, sgRNA, and ssDNA templates.

**Genotyping primers**

| Allele | Forward | Reverse | Product (bp) |
|---|---|---|---|
| *Trip13* WT | GCCCTTAGCCAAGGTGGAT | TCCTTGCACCCCTAATTGAC | 645 |
| *Trip13* KO | ACTTGCTTTAAAAAACCTCCCACA | CAGAAAGCAACTGCTCCCTTCTAGC | 731 |
| *Sycp2* WT | AGATGAGGGCATATCACCGA | TAAGCACACTCACCATCTCC | 400 |
| *Sycp2* KO | GCATGTTATCAACCTTATCCCT | CCTACCGGTGGATGTGGAATGTGTG | 300 |
| *Rec8* WT | AGCAGAGTCGAAGAAGGCCTCTTG | CAGATGGTGGCGAAGCAGCCTGT | 426 |
| *Rec8* KO | AGCAGAGTCGAAGAAGGCCTCTTG | TTGCTCAGGGGAATTTGGGTC | 212 |
| *Hormad1* WT | TCAAGACCAACCTGGGCTAC | CCATGTGGGTTGTAGGGAGT | 196 |
| *Hormad1* KO | TCAAGACCAACCTGGGCTAC | GGGGAACTTCCTGACTAGGG | ~400 |
| 3×FLAG-*Trip13* | CCTACATCGGAGAAGGCTGT | TTCATGTCAGGCTGTTCAGG | WT:348 KI:426 |
| *Trip13*-3×FLAG | GCCCCACTAAAGCACAAGTC | ACAGGCTTGAGTCAGGATGG | WT:405 KI:471 |

**Genome editing**

| | Guide RNA | HDR ssDNA |
|---|---|---|
| 3×FLAG-*Trip13* | ATTCCCTCGGCTCCCGGCGG | TCGGAGAAGGCTGTCGCACAGGGCGCAGGGA GGCGACCGCGGCCTCACTCCGGCGGCATTCC CTCGGCTCCCGGCGGCAGCGCCATGGGTGAC TACAAAGACCATGACGGTGATTATAAAGATCATG ACATCGATTACAAGGATGACGATGACAAGGGAA GCGGAGACGAGGCGGTGGGCGACCTGAAGCAAGCGCTTCC |
| *Trip13*-3×FLAG | AAGCCATAGATATGGATGTC | AGGGTTTCCTCCAGGCCCTATCTCTGGCAGTGGAC AAACAGTTTGAGGAGAAAAAGAAACTTTCAGCTTAT GTTGACTACAAAGACCATGACGGTGATTATAAAGATC ATGACATCGATTACAAGGATGACGATGACAAGTGATC CAAGACATCCATATCTATGGCTTTCAATGGACAAGTAGGAGGTGATACCGTCTAC |

## Histological, immunofluorescence, and surface nuclear spread analyses

For histology, testes or ovaries were fixed in Bouin's solution at room temperature overnight, embedded with paraffin, and sectioned at 8 μm. Sections were stained with hematoxylin and eosin. For immunofluorescence analysis, testes were fixed in 4% paraformaldehyde (in 1× PBS) overnight at 4°C, dehydrated in 30% sucrose (in 1× PBS) overnight, and sectioned at 8 μm in a cryostat. Surface nuclear spread analysis was described before (*Peters et al., 1997*). Briefly, testicular tubules or ovarian tissues were soaked in hypotonic treatment buffer (30 mM Tris, 50 mM sucrose, 17 mM trisodium citrate dihydrate, 5 mM EDTA, 0.5 mM DTT, 1 mM PMSF). Then the cells were suspended in 100 mM sucrose and spread by physical disruption on PTFE printed slides that were previously soaked with paraformaldehyde solution containing Triton X-100 and sodium borate.

## Imaging

Histological images were captured on the Leica DM5500B microscope with a DFC450 digital camera (Leica Microsystems, Wetzlar, Germany). Most immunolabeled chromosome spread images were taken on the Leica DM5500B microscope with an ORCA Flash4.0 digital monochrome camera (Hamamatsu Photonics, Bridgewater, NJ). Confocal microscopy of immunolabeled chromosome spreads was performed on a Leica SP5 II confocal (Leica Microsystems) with an ×100 (1.46 NA) oil immersion objective lens. Images were deconvolved with Huygens Essential deconvolution software (Scientific Volume Imaging B.V., Hilversum, Netherlands).

## Western blot analysis

Testes were homogenized in the lysis buffer (50 mM Tris-HCl, pH 8.0, 150 mM NaCl, 1% Triton X-100, 0.5% sodium deoxycholate, 5 mM $MgCl_2$, and 1 mM DTT supplemented with 1 mM PMSF). 40 μg

of protein samples were resolved by SDS-PAGE, transferred onto PDVF membranes, and immuno-blotted with primary antibodies.

## Acknowledgements

We thank Gordon Ruthel at the PennVet Imaging Core for help with super-resolution microscopy, Hsin Yao Tang and Thomas Beer at Wistar Proteomics Core for help with mass spectrometry, Christer Hoog for SYCP1 antibody, Attila Toth for HORMAD2 antibody, Scott Keeney for anti-REC114 antibody, Mengcheng Luo for REC8 and SYCE1 antibodies, and Yoshinori Watanabe and Takashi Akera for CENP-C antibody.This work was supported by the National Institutes of Health/National Institute of Child Health and Human Development grants T32HD083185 (JYC) and National Institutes of Health/National Institute of General Medical Sciences R35GM153384 (PJW).

## Additional information

### Funding

| Funder | Grant reference number | Author |
|---|---|---|
| Eunice Kennedy Shriver National Institute of Child Health and Human Development | T32HD083185 | Jessica Y Chotiner |
| National Institute of General Medical Sciences | R35GM153384 | P Jeremy Wang |

The funders had no role in study design, data collection and interpretation, or the decision to submit the work for publication.

### Author contributions

Jessica Y Chotiner, Conceptualization, Formal analysis, Funding acquisition, Validation, Investigation, Visualization, Methodology, Writing – original draft; N Adrian Leu, Methodology; Fang Yang, Isabella G Cossu, Investigation; Yongjuan Guan, Huijuan Lin, Resources, Investigation; P Jeremy Wang, Conceptualization, Formal analysis, Supervision, Funding acquisition, Writing – original draft, Project administration, Writing – review and editing

### Author ORCIDs

P Jeremy Wang https://orcid.org/0000-0003-2311-4089

### Ethics

Mice were maintained and used for experimentation according to the protocol (IACUC 804421) approved by the Institutional Animal Care and Use Committee of the University of Pennsylvania.

Reviewer #1 (Public Review): https://doi.org/10.7554/eLife.92195.3.sa1
Reviewer #2 (Public Review): https://doi.org/10.7554/eLife.92195.3.sa2
Author response https://doi.org/10.7554/eLife.92195.3.sa3

## Additional files

### Supplementary files

- MDAR checklist

### Data availability

All data generated or analysed during this study are included in the manuscript and supporting files.

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
