## [Editor Report · eLife assessment]

This **important** study defined the physiological function of a conserved meiosis factor during murine spermatogenesis. The genetic and cellular biological evidence supporting the conclusion is **convincing**. This work will be of broad interest to cell biologists, geneticists, and reproductive biologists.

---

## [Referee Report · Reviewer #1 (Public Review)]

Summary:

TRIP13/Pch2 is a conserved essential regulator of meiotic recombination from yeast to humans. In this manuscript, the authors generated TRIP13 null mice and Flag-tagged TRIP13 knock-in mice to study its role in meiosis. They demonstrate that TRIP13 regulates MORMA domain proteins and is essential for meiotic completion and fertility. The main impact of this manuscript is its clarification of the in vivo function of TRIP13 during mouse meiosis and previously unrecognized role as a dose-sensitive regulator of meiosis.

Strengths:

Two previously reported Trip13 mutations in mice are both hypomorphic alleles with distinct phenotypes, precluding a conclusion on its function. This study for the first time generated the TRIP13 null mice, definitively revealed the function of TRIP13 in meiosis. The authors also show novel localization of TRIP13 at SC and its independence from the axial element components. The finding of dose-sensitive regulation of meiosis by TRIP13 has implication in understanding human meiosis and disease phenotypes.

The results support the main conclusions and advance the understand of meiosis in the germline.

---

## [Referee Report · Reviewer #2 (Public Review)]

Summary and Strengths:

In this manuscript, Chotiner and colleagues demonstrated the localization of TRIP13 and clarified the phenotypes of Trip13-null mice in mouse meiosis. The meiotic phenotypes of Trip13 have been well characterized using the hypomorph alleles in the literature. However, the null phenotypes have not been examined, and the localization of TRIP13 was not clearly demonstrated. The study fills these important knowledge gaps in the field. The demonstration of TRIP13 localization to SC in mice provides an explanation of how HOMRA domain proteins are evicted from SC in diverse organisms. This conclusion was confirmed in both IF and TRIP13-tagged Tg mice. Further, the phenotypes of Trip13-null mice are very clear. The manuscript is well crafted, and the discussion section is well organized and comprehends the topic in the field. All in all, the manuscript will provide important knowledge in the field of meiosis.

Weaknesses:

The heterozygous phenotypes demonstrate that TRIP13 is a dosage-sensitive regulator of meiosis. In relation to this conclusion, as summarized in the discussion section, other mutants defective in meiotic recombination showed dosage-sensitive phenotypes. However, the authors did not examine meiotic recombination in the Trip13-null mice.

---

## [Author Response]

The following is the authors’ response to the original reviews.

**Public Reviews:**

**Reviewer #1 (Public Review):**
Summary:TRIP13/Pch2 is a conserved essential regulator of meiotic recombination from yeast to humans. In this manuscript, the authors generated TRIP13 null mice and Flag-tagged TRIP13 knock-in mice to study its role in meiosis. They demonstrate that TRIP13 regulates MORMA domain proteins and is essential for meiotic completion and fertility. The main impact of this manuscript is its clarification of the in vivo function of TRIP13 during mouse meiosis and its previously unrecognized role as a dose-sensitive regulator of meiosis.Strengths:Two previously reported Trip13 mutations in mice are both hypomorphic alleles with distinct phenotypes, precluding a conclusion on its function. This study for the first time generated the TRIP13 null mice, definitively revealing the function of TRIP13 in meiosis. The authors also show the novel localization of TRIP13 at SC and its independence from the axial element components. The finding of dose-sensitive regulation of meiosis by TRIP13 has implications in understanding human meiosis and disease phenotypes.Weaknesses:This manuscript would be more impactful if more mechanistic advancements could be made. For example, the authors could follow up with one of the new interactors identified by MS to offer new insight into the molecular function of TRIP13.

We agree that it would be interesting to follow up on new candidate interactors but think that it would be more feasible to follow up on them in future studies.

**Reviewer #2 (Public Review):**
Summary and Strengths:In this manuscript, Chotiner and colleagues demonstrated the localization of TRIP13 and clarified the phenotypes of Trip13-null mice in mouse meiosis. The meiotic phenotypes of Trip13 have been well characterized using the hypomorph alleles in the literature. However, the null phenotypes have not been examined, and the localization of TRIP13 was not clearly demonstrated. The study fills these important knowledge gaps in the field. The demonstration of TRIP13 localization to SC in mice provides an explanation of how HOMRA domain proteins are evicted from SC in diverse organisms. This conclusion was confirmed in both IF and TRIP13-tagged Tg mice. Further, the phenotypes of Trip13-null mice are very clear. The manuscript is well crafted, and the discussion section is well organized and comprehends the topic in the field. All in all, the manuscript will provide important knowledge in the field of meiosis.Weaknesses:The heterozygous phenotypes demonstrate that TRIP13 is a dosage-sensitive regulator of meiosis. In relation to this conclusion, as summarized in the discussion section, other mutants defective in meiotic recombination showed dosage-sensitive phenotypes. However, the authors did not examine meiotic recombination in the Trip13-null mice.

Meiotic recombination was extensively characterized in Trip13 severe hypomorph mutants in two previous studies: gamma-H2AX, BLM, BRCA1, ATR, RPA, RAD51, DMC1, MLH1 (Li and Schimenti, 2007; Roig et al., 2010). All the meiotic defects in our Trip13-null mice were also present in Trip13 severe hypermorph mutants: meiotic arrest, defects in chromosomal synapsis, asynapsis at chromosomal ends, and accumulation of HORMAD1/2 on the SC axis. Therefore, the defects in meiotic recombination in Trip13-null mice are expected to be similar to those in Trip13 severe hypermorph mutants and thus we did not examine the proteins involved in meiotic recombination in the Trip13-null mutant.

**Reviewer #3 (Public Review):**
Summary:The authors perform a thorough examination of the phenotypes of a newly generated Trip13 null allele in mice, noting defects in chromosome synapsis and impact on localization of other key proteins (namely HORMADs) on meiotic chromosomes. The vast majority of data confirms observations of several prior studies of Trip13 alleles (moderate and severe hypomorphs). The original or primary aims of the study aren't clear, but it can be assumed that the authors wanted to better study the role of this protein in evicting HORMADs upon synapsis by studying phenotypes of mutants and better characterizing TRIP13 localization data (which they find localizes to the central element of synapsed chromosomes using a new epitope-tagged allele). Their data confirm prior reports and are consistent with localization data of the orthologous Pch2 protein in many other organisms.Strengths:The quality of data is high. Probably the most important data the authors find is that TRIP13 is localized along the CE of synapsed chromosomes. However, this was not unexpected because PCH2 is also similarly localized. Also, the authors use a clear null (deletion allele), whereas prior studies used hypomorphs.Weaknesses:There is limited new data; most are confirmatory or expected (i.e., SC localization), and thus the impact of this report is not high. The claim that TRIP13 "functions as a dosage-sensitive regulator of meiosis" is exaggerated in my opinion. Indeed, the authors make the observation that hets have a phenotype, but numerous genes have haploinsufficient phenotypes. In my opinion, it is a leap to extrapolate this to infer that TRIP13 is a "regulator" of meiosis. What is the definition of a meiosis regulator? Is it at the apex of the meiosis process, or is it a crucial cog of any aspect of meiosis?

TRIP13 is not haploinsufficient, as Trip13 heterozygotes were still viable and fertile (albeit with defects in meiosis). TRIP13 is an ATPase and changes the conformation of meiosis-specific proteins such as HORMAD proteins. TRIP13 is essential for meiosis and its mutations cause defects in both meiotic recombination and chromosomal synapsis. Reviewer 1 stated that “TRIP13/Pch2 is a conserved essential regulator of meiotic recombination from yeast to humans”. Therefore, we feel that TRIP13 can be called a regulator of meiosis.

**Reviewer #1 (Recommendations For The Authors):**
A schematic illustration of SC structure, the components involved, and the main finding, would be helpful for readers to better understand the advancement made by this study.

We have now added a schematic illustration in a new panel - Figure 7C.

Fig. 1B, the stage with diplotene cells should be XII.

The pachytene cells (Pac) were mis-labelled as diplotene cells. Corrected.

Fig. 1C, color mislabeled.

Corrected.

**Reviewer #2 (Recommendations For The Authors):**
The manuscript will provide important knowledge in the field of meiosis. I support the publication of this study. I have some suggestions to improve and polish the manuscript.Major points:(1) The heterozygous phenotypes demonstrate that TRIP13 is a dosage-sensitive regulator of meiosis. In relation to this conclusion, as summarized in the discussion section, other mutants defective in meiotic recombination showed dosage-sensitive phenotypes. Given the function of HORMAD1 in meiotic recombination, it would be informative if the authors could examine how major makers of meiotic recombination behave in Trip13-null meiosis.

Please see our response to Weaknesses from Reviewer #2.

(2) Relating to the above point, the complete lack of synapsis on the sex chromosomes in the Trip13-null meiosis is impressive. This result raises a question as to whether the pathway to designate XY-obligatory crossover (which can be detected with large foci of ANKRD31 and MEI4/REC114 at PAR) is affected or not. It would be interesting to examine whether the ANKRD31 and MEI4/REC114 foci are present on PAR in Trip13-null meiosis.

We have performed immunofluorescent analysis of REC114 in spermatocytes. In Trip13-null pachytene-like spermatocytes, X and Y chromosomes are not synapsed. REC114 still formed one focus each on the unsynapsed X and Y chromosomes. We have added this new data in the Results as a new supplementary figure (Figure 4 -supplement 1).

(3) Figure 4 can be improved if there are quantified data for each phenotype. These phenotypes look nearly complete, but it would be informative to show the penetrance of these phenotypes.

Because some chromosomes have unsynapsed ends, resulting in two centromere or telomere foci, the total number of centromere or telomere foci is always higher in Trip13-null pachytene-like spermatocytes than wild type pachytene spermatocytes. Therefore, we did not count the foci of centromeres and telomeres. Consistently, the centromere and telomere markers localized as expected in both wild type and Trip13-null spermatocytes.

(4) I am not fully convinced by these photos: "synapsed sister chromatids (Figure 6B)" and "Sycp2-/- spermatocytes formed short stretches of synapsis (Figure 6C)". The authors may try confocal microscopy with super-resolution deconvolution as they did for other data.

These have been previously demonstrated. The “synapsed sister chromatids (Figure 6B)” were previously demonstrated by confocal microscopy with super-resolution deconvolution (Guan et al., 2020). The short stretches of synapsis in Sycp2-/- spermatocytes was previously demonstrated by electron microscopy (Tripartite SC structure) and SYCP1 immunofluorescence (Yang et al., 2006). We have revised the text by citing the previous evidence and the publications.

Minor points:(1) Line 19-21: "Loss of TRIP13 leads to meiotic arrest and thus sterility in both sexes. Trip13-null meiocytes exhibit abnormal persistence of HORMAD1 and HOMRAD2 on synapsed SC". These findings confirm the previously reported phenotypes of the Trip13 hypomorph alleles. This information can be added to the abstract. Otherwise, it sounds like these are totally new findings, as written.

This information is now added to the abstract: “These findings confirm the previously reported phenotypes of the Trip13 hypomorph alleles.”

(2) The introduction section seems too long and contains unnecessary information. Some molecular details that are not touched in the result section can be deleted (e.g., Line 65-73).

We would like to keep the molecular details on the two conformation states, as it provides biochemical background on TRIP13-HORMAD interactions.

(3) Introduction, Line 92. A rationale can be added as to why the authors characterized the Trip13-null allele.

a rationale has been added as follows: “To determine the effect of complete loss of TRIP13, we characterized Trip13-null mice.”

(4) Line 205: Typo "TRRIP13".Corrected.
**Reviewer #3 (Recommendations For The Authors):**
Just a few recommendations:(1) In my opinion, the title is an overreach. "Regulator" invokes other concepts such as transcription factors.

Please see our explanation in response to weaknesses from Reviewer #3.

(2) The first sentence of the results deals with TRIP13 expression in only 3 tissues. The authors might look at more comprehensive RNA-seq data from mice and humans.

We examined TRIP13 protein expression in 8 mouse tissues by WB and found that TRIP13 protein was abundant in testis but present at a very low level in ovary and liver (Figure 1A). We feel that readers can easily look up the relative transcript levels of Trip13 in more tissues from mice and humans from NCBI database under “Gene”.

(3) The null allele is semi-lethal. Is body size affected? Were the mice abnormal in any other ways, given that TRIP13 has been implicated in other diseases and processes, and is expressed in other tissues (TRIP13 stands for Thyroid receptor interacting protein).

The body weight of 2-3 month-old males was not significantly different between wild type (24.3±2.8 g, n=5) and Trip13 KO mice (22.8±1.7 g, n=5, p=0.3, Student’s t-Test). We have included the body weight information in the revised manuscript. We didn’t observe abnormal somatic defects in the viable Trip13-null mice, nor did the authors report any in the Trip13 hypomorph mutants in two previous studies (Li and Schimenti, 2007; Roig et al., 2010).

(4) Line 276 : It would be nice to elaborate on the "spatial explanation."

We meant that TRIP13 localizes to SC while HORMAD proteins are removed from SC upon chromosomal synapsis, thus providing a spatial explanation. However, we have now deleted “spatial”.